# Exploration of Genes Related to Intramuscular Fat Deposition in Xinjiang Brown Cattle

**DOI:** 10.3390/genes15091121

**Published:** 2024-08-25

**Authors:** Yu Gao, Liang Yang, Kangyu Yao, Yiran Wang, Wei Shao, Min Yang, Xinyu Zhang, Yong Wei, Wanping Ren

**Affiliations:** Xinjiang Key Laboratory of Meat and Milk Production Herbivore Nutrition, College of Animal Science, Xinjiang Agricultural University, Urumqi 830052, China; yugao12123@163.com (Y.G.); yangliangagu@sina.com (L.Y.); kangyuyao930@163.com (K.Y.); wyr2107937740@163.com (Y.W.); dksw@xjau.edu.cn (W.S.); y1375724150@163.com (M.Y.); 18663955139@163.com (X.Z.); wy-260@163.com (Y.W.)

**Keywords:** longissimus dorsi muscle, intramuscular fat, differentially expressed gene, transcriptome sequencing technology, metabolic pathway

## Abstract

The aim of this study was to investigate the differentially expressed genes associated with intramuscular fat deposition in the longissimus dorsi muscle of Xinjiang Brown Bulls. The longissimus dorsi muscles of 10 Xinjiang Brown Bulls were selected under the same feeding conditions. The intramuscular fat content of muscle samples was determined by the Soxhlet extraction method, for which 5 samples with high intramuscular fat content (HIMF group) and 5 samples with low intramuscular fat content (LIMF group) were selected. It was found that the intramuscular fat content of the HIMF group was 46.054% higher than that of the LIMF group. Muscle samples produced by paraffin sectioning were selected for morphological observation. It was found that the fat richness of the HIMF group was better than that of the LIMF group. Transcriptome sequencing technology was used to analyze the gene expression differences of longissimus dorsi muscle. Through in-depth analysis of the longissimus dorsi muscle by transcriptome sequencing technology, we screened a total of 165 differentially expressed genes. The results of Gene Ontology (GO) enrichment analysis showed that the differentially expressed genes in the two groups were mainly clustered in biological pathways related to carbohydrate metabolic processes, redox processes and oxidoreductase activities. Kyoto Encyclopedia of Genes and Genomes (KEGG) enrichment analysis showed that the differentially expressed genes were significantly clustered in 15 metabolic pathways, which mainly covered fatty acid metabolism (related to lipid metabolism and glucose metabolism), the pentose phosphate pathway, the Peroxisome Proliferator-Activated Receptor (PPAR) signaling pathway and other important metabolic processes. The three genes that were predominantly enriched in the glycolipid metabolic pathway by analysis were *SCD5*, *CPT1C* and *FBP2*, all of which directly or indirectly affect intramuscular fat deposition. In summary, the present study investigated the differences in gene expression between high and low intramuscular fat content in the longissimus dorsi muscle of Xinjiang Brown Bulls by transcriptome sequencing technology and revealed the related signaling pathways. Therefore, we hypothesized that SCD5, CPT1C and FBP2 were the key genes responsible for the significant differences in intramuscular fat content of the longissimus dorsi muscles in a population of Xinjiang Brown Bulls. We expect that these findings will provide fundamental support for subsequent studies exploring key genes affecting fat deposition characteristics in Xinjiang Brown Bulls.

## 1. Introduction

Xinjiang Brown Cattle, a dairy–beef dual-purpose breed from Xinjiang, China, was cultivated and bred independently in China. Xinjiang Brown Cattle inherited the strong adaptability of female Kazakh Cattle (heritability 9.88%) and the better production performance (beef and dairy) of male Swiss Brown Cattle (heritability 90.12%) [1]. At present, through nutritional regulation and other technical means, the carcass quality grade of Xinjiang Brown Cattle has generally reached the level of above R grade (referring to the European Union (EU) beef cattle carcass quality grade standard) [2,3]. Previous studies have shown that in Xinjiang Brown Steers (10–12 months of age), the ribeye area was 106.06 cm^2^, the fat cover was 3.5%, the marbling score was grade 3 to 4 and the fat color score was grade 2 (referring to NY/T 676-2010 Beef Grade Specification) [3]. Therefore, Xinjiang Brown Cattle can be the main target of high-grade beef production.

Intramuscular fat (IMF) is mainly distributed in the outer and inner muscle membranes as well as the fascia of the muscle, which has an important impact on the flavor, tenderness and nutritional value of beef [4]. Yan et al. [2] reported that the IMF content of Xinjiang Brown Steers (6.96–7.50%) was higher than that of Angus Steers (6.2%). IMF deposition is mainly regulated by genes related to lipid metabolism. Lipid metabolism is a complex dynamic equilibrium involving the synthesis and catabolism of adipocytes, which is enabled by the co-regulation of multiple transcription factors [5]. Studies have shown that differences in key lipid-controlling genes in the beef cattle organism are a key factor contributing to individual differences in intramuscular fat deposition capacity between breeds [6]. Therefore, exploring the main effector genes affecting IMF deposition in order to clarify the production and regulation mechanism of IMF deposition in beef cattle is an effective way to improve IMF content in beef.

With RNA-Seq (transcriptome sequencing) technology, it is possible to accurately capture and resolve the expression of all transcription products within a species under specific conditions [7]. The technology is capable of quantifying mRNA expression levels by means of high-throughput sequencing, and it can be further combined with bioinformatics analysis methods to study the characteristics of genes at the transcriptional level in depth, including the amount of gene expression as well as the differences in the expression patterns among different samples. RNA-Seq technology is capable of achieving read lengths of 2 × 150 bp, and it has the advantage of high sequencing throughput and high accuracy (>99%) for precise quantification [8].

RNA-Seq technology can be used to study the differentially expressed genes related to the characteristics of muscle fatty acid composition and intramuscular fat content [9,10]. Cesar AS et al. [10] succeeded in revealing molecular mechanisms closely related to lipid metabolism by performing tissue transcriptome (mRNA sequencing) analyses and identified differentially expressed genes present within muscle tissues in Nile beef cattle populations. Zheng et al. [6] sequenced the transcriptome of longissimus dorsi muscle tissue samples from Angus and Chinese Simmental cattle and identified 17 genes that are closely related to muscle and fat metabolism. However, it is still unclear as to which genes regulate intramuscular fat deposition differences among individuals of Xinjiang Brown Cattle. Therefore, we hypothesized that certain genes affect intramuscular fat deposition in Xinjiang Brown Cattle. We used transcriptome sequencing to analyze the differentially expressed genes in the longissimus dorsi muscle of Xinjiang Brown Cattle and explored the genes and pathways closely related to intramuscular fat deposition in conjunction with existing studies to gain insight into the key genes that regulate intramuscular fat deposition.

## 2. Materials and Methods

### 2.1. Animals

In this experiment, 10 healthy 26-month-old Xinjiang Brown Bulls (653.21 ± 67.27 kg) with the same genetic background, kept under the same feeding conditions, were randomly selected at the Xinjiang Yili New Brown Breeding Farm. The diets were formulated according to the NRC (National Research Council) Nutrient Requirements for Beef Cattle and included whole plant silage corn, corn stover, alfalfa, wheat straw, corn kernels, cottonseed meal, soybean meal and bran. This study was approved by the Animal Ethics Committee of Xinjiang Agricultural University.

### 2.2. Sample Collection

All Xinjiang Brown Bulls were humanely slaughtered at the Shengyuan Cattle and Sheep Designated Slaughterhouse in Changji, Xinjiang. Longissimus dorsi muscle samples were collected immediately, cut into small pieces using surgical scissors carefully, and then quickly placed into freezing tubes that were put into liquid nitrogen for cryopreservation. We determined the intramuscular fat (IMF) content of the longissimus dorsi muscle of 10 Xinjiang Brown Bulls using the Soxhlet extraction method and subsequently ranked the samples according to the measured content from highest to lowest. On this basis, the top five samples with the highest IMF content were selected as the HIMF group, while the remaining samples were categorized as the LMIF group. At the same time, we divided the muscle samples in both the HIMF and LIMF groups into two parts. One part of the samples (20–50 g) was immersed in 4% formalin to make paraffin sections for morphological observation, and the other (2–4 g) was frozen in liquid nitrogen for transcriptome sequencing technology to analyze the gene expression differences of the longissimus dorsi muscle between the two groups.

### 2.3. Determination of IMF Content

IMF content was determined by Soxhlet extraction. A 3 g sample of the longissimus dorsi muscle was stirred into a dried paper packet and weighed. After continuous drying in an oven at 65 °C for 15 h or more, the paper packets were then placed in a desiccator to cool for 10 min and subsequently weighed. Subsequently, the paper packets were transferred to a Soxhlet extractor and soaked in ether overnight. Afterward, they were refluxed in a water bath heated to 75 °C for 10 h and then placed in a well-ventilated area to allow the ether to evaporate naturally. Finally, the samples were dried in an oven at 105 °C until their weight no longer changed, i.e., a constant weight state was reached, and subsequently weighed, from which the IMF content was calculated.

### 2.4. Histology

Five samples of the longissimus dorsi muscle were selected from each of the HIMF and LIMF groups and subsequently cut along the natural texture of the muscle fibers into muscle tissue blocks, each measuring 1 cm × 1 cm × 0.5 cm. The tissues were immersed in 4% paraformaldehyde fixative (Wuhan Carnoss Technology Co., Ltd., Wuhan, China) for a period of 24 h to complete the tissue fixation process. They were then placed in a series of alcohol (Xinjiang Aidil Biotechnology Co., Ltd., Urumqi, China) solutions at increasingly concentrated levels to progressively remove water from the tissue samples. Finally, the tissue samples were transferred to xylene (Heng Xing Reagent, Tianjing, China) for hyalinization. The samples were immersed in a paraffin (Shanghai Yi Yang Instrument Co., Shanghai, China) solution for the purpose of embedding, after which they were left to cool and solidify naturally. Subsequently, they were sectioned. The cut paraffin sections were placed on slides and deparaffinized using xylene (Heng Xing Reagent, Tianjing, China). Next, the samples were stained with hematoxylin (Zhuhai Beso Biotech Co., Ltd., Zhuhai, China) and slightly washed with running water. They were then differentiated with 0.1% hydrochloric acid (concentrated hydrochloric acid/anhydrous ethanol = 1:1) in ethanol and washed with water. Subsequently, the sections were subjected to staining with eosin (Zhuhai Beso Biotech Co., Ltd., Zhuhai, China), followed by a washing and dehydration step involving alcohol. Ultimately, the slices were treated with xylene transparency and sealed with resin (Shanghai Yi Yang Instrument Co., Shanghai, China), thus completing the preparation process. The tissue sections were then examined in morphological detail using a FEI Quanta 250 Field Emission Environmental Scanning Electron Microscope (FEI Corporation, Hillsboro, OR, USA).

### 2.5. Total RNA Extraction, Library Construction and Sequencing

The muscle samples were carefully ground into powder form, from which total RNA was extracted using TRIzol reagent (Beijing Adderall Biotechnology Co., Ltd., Beijing, China). mRNA molecules with polyA tails were specifically enriched using Oligo(dT) magnetic bead technology. The integrity of mRNA was accurately examined using an Agilent 2100 Bioanalyzer (Agilent Technologies, Santa Clara, CA, USA).

A library was constructed using 5 μg of RNA, and the first strand of cDNA was synthesized using fragmented mRNA as a template and random oligonucleotides (Guangzhou Reebok Biotechnology Co., Guangzhou, China) as primers in the M-MuLV reverse transcriptase system. The RNA strand was degraded using RNaseH (Shanghai Lianmai Biological Engineering Co., Shanghai, China), and the second strand of cDNA was subsequently synthesized using dNTPs in a DNA polymerase I system. The purified double-stranded cDNA underwent end repair, the addition of an A-tail and ligation of sequencing junctions. Afterward, PCR amplification was performed, and the PCR products were purified again using AMPure XP beads (Beckman Coulter, Brea, CA, USA) to produce the final library.

The libraries were initially quantified using a Qubit 2.0 Fluorometer (Thermo Fisher Scientific, Waltham, MA, USA) and subsequently diluted to a concentration of 1.5 ng/μL. Next, the insert size of the libraries was examined using an Agilent 2100 bioanalyzer. After it was confirmed that the insert sizes were as expected, the libraries were pooled according to the effective concentration of each library and the target sequencing data volume for Illumina sequencing. In the flow cell for sequencing, we added four kinds of fluorescently labeled dNTP, DNA polymerase and junction primers for amplification. The sequencer captured these fluorescent signals and converted them into sequencing peaks through the computer software to obtain the sequence information of the DNA fragments to be sequenced.

### 2.6. Real-Time Quantitative PCR (RT-qPCR)

RNA samples from transcriptome sequencing were taken for RT-qPCR validation with three biological replicates for each sample, and cDNA was obtained by operating according to the instructions of the Reverse Transcription Kit (Nanjing Novozymes Biotechnology Co., Ltd., Nanjing, China). Data on 10 differential genes associated with fat metabolism among individuals of Xinjiang Brown Cattle were identified using an ABI SteponePlus instrument (Applied Biosystems, Inc., Waltham, MA, United States) according to the protocol outlined in the Fluorescence Quantification Kit for Genetically Modified Organisms Technology (Nanjing Novozymes Biotechnology Co., Ltd., Nanjing, China). Primer sequences were designed using Primer 5 (Table 1), and the internal reference gene was *β-actin*. The RT-qPCR amplification program was set to 95 °C for 5 min, 95 °C for 10 s and 60 °C for 30 s (40 cycles), and the results were analyzed by the 2^−ΔΔCT^ method.

### 2.7. Data Processing

#### 2.7.1. Quality Control, Transcript Assembly and Splicing

Sequenced fragments were converted into reads by CASAVA base recognition of the image data measured by the high-throughput sequencer, and the files were in fastq format. Filtering from raw read length to net read length was performed. Filtering content was as follows: removal of reads with adapters (connectors), removal of reads containing N (N indicates that base information could not be determined) and removal of low-quality reads (reads where the number of bases with Qphred ≤ 5 accounted for more than 50% of the entire read length). Q20, Q30 and GC content were calculated for the net data. The reference genome and gene annotation file (Bos_taurus.ARS-UCD1.2) were downloaded from the ensemb database, and the clean reads were quickly and accurately compared with the reference genome using HISAT2 (version 2.0.5) software. Based on the comparison information, StringTie (version v2.2.1) software was applied to assemble the new transcripts.

#### 2.7.2. Sequence Data Mining and Analysis

Expression levels of genes were expressed as transcript fragments per kilobase per million mapped read values. Differentially expressed mRNAs were obtained from the expression level analysis of mRNAs, and transcripts were analyzed for differences using the negative binomial distribution of DESeq2 (1.20.0). *p*-adjust ≤ 0.05 and |log2(Fold Change)| ≥ 1 were set to screen for differentially expressed genes.

#### 2.7.3. Gene Ontology (GO) and Kyoto Encyclopedia of Genes and Genomes (KEGG) Enrichment Analysis of Differentially Expressed Genes

The differential gene sets were analyzed for GO functional enrichment and KEGG pathway enrichment using clusterProfiler (version 3.8.1) software, and both GO functional enrichment and KEGG pathway enrichment were analyzed with *p* < 0.05 as the threshold for significant enrichment.

#### 2.7.4. Statistical Analysis

The data were organized using Microsoft Office Excel (version 2016, Microsoft Corporation, Redmond, WA, USA). The data of IMF content and RT-qPCR were statistically analyzed by an independent *t* test using SPSS (version 20.0, IMB Corporation, Armonk, NY, USA), and the results were expressed as mean ± standard deviation. Gene quantification was performed using the default parameters of the featureCounts (1.5.0-p3) software. R (Version 3.5.0) was used to plot heatmaps of correlation between samples (cor function package, method = person; ggplot2), box plots of the distribution of gene expression in the samples (ggplot2 package), plots of the results of the principal component analysis (ggplot2 package), and Venn diagrams of co-expression (Venn Diagram package). Difference multiplicity plots were plotted using Origin (version 7.5, OriginLab Corporation, Northampton, MA, USA).

## 3. Results

### 3.1. Comparison of IMF Content and Morphological Observation of the Longissimus Dorsi Muscle in Two Groups

The IMF content in the longissimus dorsi muscle of the 10 Xinjiang Brown Bulls in the experiment differed among individuals, with the overall IMF content in the HIMF group being 46.054% higher than that in the LIMF group (*p* = 0.002) (Table 2). Transverse sections of the longissimus dorsi muscle of both groups are shown in Figure 1. Paraffin sections were made of muscle samples from the HIMF and LIMF groups, and it was clearly observed that the texture and abundance of fat in the five sliced samples from the HIMF group were superior to those of the LIMF group.

### 3.2. Transcriptome Sequencing Data Analysis

For the longissimus dorsi muscle samples of Xinjiang Brown Bulls, an average of 556,876,666,000 raw read lengths were obtained for each sample, and an average of 43,276,318,600 filtered read lengths were obtained after raw data filtering. All of them were below 0.03% (Q20 ≥ 97.81%, Q30 ≥ 93.85%) as checked by the sequencing error rate, indicating that the sequencing data were valid (Table 3). When analyzed against the bovine reference genome (Table 4), clean reads maintained an overall match rate of more than 93.10% and a unique match rate of more than 90.04%. Only a small number of reads matched to multiple locations in the reference genome (≤3.41%), and the vast majority of reads matched uniquely to the reference genome.

### 3.3. Quantitative Analysis of Transcriptome Sequencing

According to the Venn diagram, the number of expressed genes in the HIMF group was 12625, while the number of expressed genes in the LIMF group was 11796. There were 11,486 co-expressed genes in both groups (Figure 2A). Component analysis of the 10 samples showed that both groups were clustered with good inter-sample reproducibility (Figure 2B). In terms of the distribution of gene expression in the samples, the mRNA expression levels were relatively homogeneous in the two groups of samples (Figure 2C). The correlation of gene expression levels between samples is an important indicator for testing the reliability of the experiment and whether the sample selection is reasonable. The closer the correlation coefficient is to 1, the higher the similarity of expression patterns between samples. According to the inter-sample correlation heatmap, it can be seen that the R^2^ values of the samples in this test were all greater than 0.94, indicating a high inter-sample correlation coefficient (Figure 2D).

### 3.4. Transcriptome Sequencing of Differentially Expressed Genes

A total of 165 differential genes were screened in the two groups of longissimus dorsi muscle samples, including 101 upregulated genes and 64 downregulated genes (Figure 3A). The distribution of differential genes among samples can be visualized in a volcano plot (Figure 3B). As shown by the heatmap of differentially expressed gene clustering, the two groups of differentially expressed genes were well clustered (Figure 3C).

### 3.5. GO Enrichment Analysis

The 30 most significant terms from the GO enrichment analysis results were selected to be plotted in a scatter plot for presentation (Figure 4). Among them, most of the differentially upregulated genes were enriched in 18 terms, and most of the differentially downregulated genes were enriched in 16 terms. There are four processes that are co-enriched by both: oxidation–reduction process, carbohydrate metabolic process, non-membrane-bounded organelle and oxidoreductase activity. According to Figure 4, the term involved in the oxidation–reduction reaction in biological processes was the most significant (*p* = 0.0036), followed by the term involved in carbohydrate metabolic processes (*p* = 0.0305). In terms of molecular function, the main focus is on the term of oxidoreductase activity, which includes the action of oxidoreductase activity on the aldehyde or oxygen group of the donor, the CH-OH group, the CH-OH group as a donor, and NAD or NADP as an acceptor.

### 3.6. KEGG Pathway Analysis

The 20 KEGG pathways with the most significant KEGG enrichment results were selected to be plotted in a scatter plot for presentation (Figure 5). Most of the differentially upregulated genes were highly significantly enriched in the glucagon signaling pathway (*p* = 0.0072), C-type lectin receptor signaling pathway (*p* = 0.0083), pyruvate metabolism (*p* = 0.0047) and PPAR signaling pathway (*p* = 0.0026), as well as significantly enriched in tyrosine metabolism (*p* = 0.0280), propanoate metabolism (*p* = 0.0264) and the fatty acid metabolism (*p* = 0.0118) signaling pathway. Most of the differentially downregulated genes were highly significantly enriched in glycolysis/gluconeogenesis (*p* < 0.0001) and the HIF-1 signaling pathway (*p* = 0.0018), as well as significantly enriched in fatty acid degradation (*p* = 0.0417), fructose and mannose metabolism (*p* = 0.0312), biosynthesis of amino acids (*p* = 0.0222), the pentose phosphate pathway (*p* = 0.0205) and carbon metabolism signaling pathways (*p* = 0.0129). According to Table 5, it can be seen that among the differentially upregulated genes, *SCD5* was enriched in four pathways, while *CPT1C* was enriched in six pathways, and among the differentially downregulated genes, *FBP2* was enriched in six pathways. Meanwhile, the three were co-enriched in the AMPK signaling pathway.

### 3.7. RT-qPCR Validation of RNA-Seq Results

Ten genes were screened in the transcriptome sequencing results for RT-qPCR validation, and the results showed that the transcriptome sequencing data were consistent with the data expression results of RT-qPCR (Figure 6). This proved the accuracy and reliability of the transcriptome sequencing results.

## 4. Discussion

Fat deposition represents the balance between fat synthesis and breakdown metabolism in an animal’s body, as well as the equilibrium between energy intake and expenditure. Intramuscular fat (IMF) is the result of fat deposition within the muscle tissue. Studies have demonstrated that IMF in the perimysial connective tissue enhances meat tenderness by weakening the association between collagen fibers and reducing the force required to disrupt connective tissue [11]. In our study, by measuring IMF content in the longissimus dorsi muscle of Xinjiang Brown Bulls, it was found that there were differences in IMF content between samples (4.790–6.996%). Yan et al. [2,3] found that the IMF content of Xinjiang Brown Bulls at 18 months of age ranged from 4.89% to 6.96%, and that of Xinjiang Brown Steers at 28–34 months of age ranged from 5.2% to 6.94%, which is consistent with the results of the present study.

IMF content is one of the main factors affecting the taste of beef, which is influenced by multiple factors such as genetics, nutrition and environment [12]. Screening for genes associated with meat quality traits has become increasingly common with the development of technologies for high-throughput sequencing, a process that generates massive amounts of transcriptional data, especially in the mammalian field. He et al. [13] investigated the results of gene expression in different parts of adipose tissue of Qinchuan cattle by transcriptome sequencing and obtained 5864 differentially expressed genes. Li Na [14] sequenced the transcriptome of the longest dorsal muscle of improved Xinjiang Brown Cattle and obtained a total of 1669 differentially expressed genes related to meat quality traits, including 879 upregulated genes and 790 downregulated genes. In our study, we screened a total of 165 differential genes, including 101 upregulated genes and 64 downregulated genes, in two groups of longissimus dorsi muscle samples by comparatively analyzing the transcriptome data of Xinjiang Brown Bulls’ longissimus dorsi muscles. This suggests that there are still differences in gene expression among individuals in Xinjiang Brown Bulls, and it is possible that the up- or downregulation of certain genes is a key factor affecting IMF deposition.

The analysis of biological functions and metabolic pathways helps to better analyze differentially expressed genes. In this study, GO functional enrichment analysis revealed that the differentially upregulated genes were mainly focused on redox reactions in biological processes, while at the molecular functional level, these genes were mainly focused on oxidoreductase activities. Studies have shown that a variety of physiological processes within the cells of organisms involve and depend on the conduct of various redox reactions, including energy production, signaling, enzyme-catalyzed reactions, cell proliferation and differentiation, cellular autophagy and apoptosis [15,16]. From this, it is speculated that various oxidative reactions in the organism, which work together in the process of IMF deposition, allow IMF to accumulate in muscle tissue under specific conditions.

In our study, KEGG enriched pathway analysis showed that most differentially upregulated genes were significantly enriched in the PPAR signaling pathway, pyruvate metabolism, fatty acid metabolism, fatty acid degradation and fructose and mannose metabolism pathways. Among them, the PPAR signaling pathway (including three isoforms of PPARα, PPARβ/δ and PPARγ) plays an important role in early adipogenesis by regulating fatty acid production through the regulation of phosphoenolpyruvate, while the PPARγ signaling pathway was found to exhibit high expression in adipose tissue [17]. The fatty acid metabolic pathway involves the processes of fatty acid synthesis (fatty acid synthesis pathway) as well as fatty acid β-oxidation (fatty acid degradation pathway), in which fatty acid synthesis is a dynamic balance between triglyceride catabolism and re-esterification processes [18,19]. It has been shown that triglyceride re-esterification requires the release of cytoplasmic phosphoenolpyruvate carboxylase (PEPCK-C), which is the main glycolytic enzyme of adipose tissue [20]. Mannose interferes with glucose metabolic pathways and is beneficial in preventing obesity [21]. Sharma et al. [22] found that feeding weaned mice a high-fat diet (HFD), normal diet (ND) and drinking water supplemented with mannose resulted in almost the same body weight of HFD mice weaned for 12 weeks as that of the ND mice, but found that the HFD mice that were not supplemented with mannose in the drinking water gained weight, suggesting that the addition of mannose to the mice’s drinking water could help prevent obesity. It can be hypothesized that these pathways are collectively involved in regulating fat synthesis and metabolic processes that affect IMF deposition.

IMF deposition is a complex biological process that results from the co-regulation of multiple genes. By analyzing and comparing the transcriptome data of the longissimus dorsi muscle of Xinjiang Brown Bulls, we found that the differentially upregulated genes were mainly enriched in metabolic pathways such as the PPAR signaling pathway, pyruvate metabolism and fatty acid metabolism, and the differentially downregulated genes were mainly enriched in metabolic pathways such as fatty acid degradation, glycolysis/glycogenesis, and fructose and mannose metabolism. The differentially upregulated genes *SCD5* and *CPT1C* and the differentially downregulated gene *FBP2* were mainly enriched in the metabolic pathways of the PPAR signaling pathway, pyruvate metabolism, fatty acid metabolism and fatty acid degradation. In our study, the *SCD5* gene belongs to the differentially upregulated genes, which are located in the PPAR signaling pathway and fatty acid metabolism signaling pathway. *SCD* (including *SCD1* and *SCD5* isoforms) was found to be the main enzyme regulating the conversion of saturated fatty acids to unsaturated fatty acids, which is involved in adipocyte metabolism and fatty acid metabolism [23,24], and has the function of controlling cellular fat distribution, lipid synthesis, cellular signaling, and cellular growth and replication [25,26], but there are fewer studies on the mechanisms regulating fatty acid metabolism, and fat deposition mechanisms have been less studied. Gervais R et al. [27] discovered the *SCD5* gene in the mammary gland of dairy cows and pointed out that it collaborates with *SCD1* in regulating fatty acid metabolism. Zhang HB et al. [28] found that an external spine could promote intramuscular fat deposition and increase the tenderness of muscle tissues by increasing the expression of the *SCD5* gene, thus improving the quality of the meat, but the specific regulatory mechanism was not clear. Fang QH et al. [29] knocked out the *SCD5* gene using the gene targeting knockout technology and found that the deletion of *SCD5* significantly reduced the relative content of erucic acid. Erucic acid was able to regulate MSC (mesenchymal stem cell) differentiation by inhibiting the transcriptional activity of PPARγ and significantly reduced the expression of genes related to adipocyte differentiation. Therefore, chronic intake of high erucic acid may affect lipid metabolism [30,31]. In summary, we hypothesized that the *SCD5* gene may have an active role in lipid metabolism and be involved in regulating IMF deposition.

*CPT1C* belongs to the *CPT1* family of transmembrane integrins and is associated with the transmembrane transport of mitochondrial outer membrane peptide chains [32]. It is involved in the regulation of fatty acid oxidative catabolism (FAO), which is the main source of cellular energy. In our study, the *CPT1C* gene belonged to the differentially upregulated genes, which were mainly enriched in fatty acid degradation, fatty acid metabolism, the glucagon signaling pathway and the PPAR signaling pathway. Hada T et al. [33] overexpressed three *CPT1* isoforms in COS7 cells, normalized the activity values according to the expression level of CPT1 and found that *CPT1C* had low catalytic activity, with only 2% of the *CPT1A*-specific activity and 5% of the *CPT1B*-specific activity. However, Sierra AY et al. [34] isolated microsomal cascade FAO metabolizing enzyme activity after the transfection of HEK293T and PC-12 cells using pIRES-*CPT1c* compared with cascade FAO metabolizing enzyme activity of a cascade transfected with an empty pIRES vector and found that there was a significant elevation of residual *CPT1A* activity in microsomes, while the inhibitory effect of malonyl-coenzyme A on *CPT1* was significantly reduced. Chen P et al. [35] showed that lipidomic analysis of proliferation and senescence of human embryonic lung MRC-5 fibroblasts revealed that the gain of function of *CPT1C* resulted in reduced lipid accumulation and reversal of aberrant lipid metabolism reprogramming in late-stage MRC-5 cells; by oil red O staining and a Nile red fluorescence assay, they found that when *CPT1C* function was absent, lipid accumulation was significantly reduced. Therefore, in combination with the result that *CPT1C* belongs to the differentially upregulated genes expressed in the longissimus dorsi muscle of Xinjiang Brown Bulls in the present study, it is hypothesized that *CPT1C* may indirectly enhance the ability of FAO by binding to malonyl-coenzyme A, which promotes the oxidative catabolism metabolism of fatty acids and exerts a positive effect on the deposition of IMF.

As a key enzyme in glucose metabolism, the *FBP2* gene not only regulates the expression of lipid-metabolism-related enzymes but also catalyzes the conversion of fructose 1.6-bisphosphate to fructose 6-phosphate. This conversion is essential for the synthesis of glycogen from carbohydrate precursors such as lactate. In addition, the *FBP2* gene exhibits phosphatase activity and plays a regulatory role in gluconeogenesis [36,37]. In our study, the *FBP2* gene belonged to the differentially downregulated genes, which were mainly enriched in fructose and mannose metabolism, the pentose phosphate pathway, carbon metabolism, and glycolysis/glycogenesis signaling pathways, but the *FBP2* gene belonged to the differentially upregulated genes in the AMPK signaling pathway. Ectopic expression of *FBP2* was found to activate the AMPK signaling pathway and inhibit the Akt-mTOR pathway, leading to inhibition of glucose metabolism [38]. Bakshi I et al. [39] suggested that elevated levels of *FBP2* activity may promote inefficient cycling and increase metabolic sensitivity, thereby elevating energy demand and leading to intensified substrate oxidation. Enhanced *FBP2* activity promotes glycolytic fluxes in the mouse extensor digitorum longus (EDL muscle). In addition, overexpression of *FBP2* leads to an increase in glucose oxidation in red EDL muscle while decreasing glucose oxidation in white EDL muscle, a process that may trigger differences in insulin-stimulated glucose uptake. It is hypothesized that *FBP2* may reduce unnecessary energy expenditure during sugar metabolism and help achieve an optimal state of glycolytic efficiency. Rukkwamsuk T et al. [40] found that the rate of gluconeogenesis was slowed down and thus triggered a delay in lipolysis due to a decrease in FBP2 activity in cows with steatohepatitis during the postpartum period of 1–7 days, resulting in an impaired gluconeogenic capacity of the liver. It can be speculated that this gene may play a role in delaying lipolysis in muscle tissue, thus promoting a more abundant intramuscular fat content in the LIMF group.

This paper analyzed the relevant pathways and differentially expressed genes that may affect IMF deposition in Xinjiang Brown Bulls with the help of transcriptome sequencing technology. We found that the differentially upregulated genes *SCD5* and *CPT1C* were co-enriched in fatty acid metabolism and the PPAR signaling pathway, the differentially downregulated gene *FBP2* was mainly enriched in glycolysis/glycogenesis and fructose and mannose metabolism, and the three genes were co-enriched in the AMPK signaling pathway. Based on the above discussion, we hypothesized that the *SCD5*, *CPT1C* and *FBP2* genes may directly or indirectly affect IMF deposition through certain signaling pathways, leading to the production of differences in IMF content among individuals of Xinjiang Brown Bulls. However, this study did not validate these three genes in cellular assays, but only made relevant speculations based on previous studies. Our team expects to use the results of this study as a follow-up to explore the key genes affecting IMF deposition, in the hope of identifying the molecular mechanisms affecting IMF content, an important meat quality trait, in Xinjiang Brown Bulls.

## 5. Conclusions

Overall, our study indicates that the *SCD5* and *CPT1C* genes (related to fatty acid metabolism and the PPAR signaling pathway) can indirectly or directly promote intramuscular fat deposition, while the *FBP2* gene (involved in glycolysis/glycogenesis and AMPK signaling pathway) can influence adipose lipid metabolism through glucose metabolism, suggesting that *SCD5*, *CPT1C* and *FBP2* are differentially expressed genes that participate in regulating lipid-metabolism-related processes. Therefore, it is necessary to further investigate the potential mechanisms by which the *SCD5*, *CPT1C* and *FBP2* genes regulate intramuscular lipid content in the longissimus dorsi muscle of Xinjiang Brown Cattle. Despite the limitations of this study, these results provide a reference for exploring the key regulatory genes affecting the fat deposition process in cattle.

## Figures and Tables

**Figure 1 genes-15-01121-f001:**
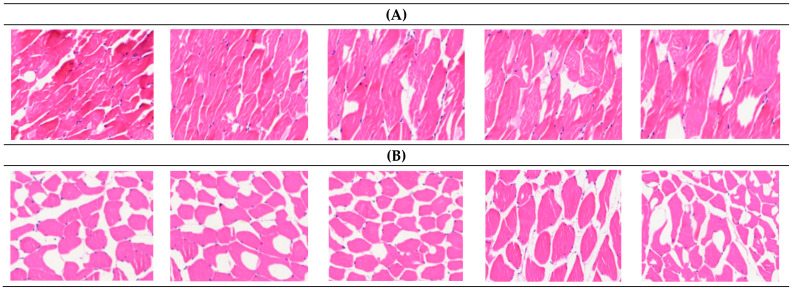
Morphological observation on the longissimus dorsi muscle of Xinjiang Brown Bulls. (**A**) Paraffin sectioning of 5 samples of intramuscular fat content in the LIMF group; (**B**) paraffin sectioning was performed on five samples of intramuscular adiposity in the HIMF group. Hematoxylin eosin staining was used, the nucleus was dyed blue, muscle fibers were dyed red, fat tissue was white, and the adipose tissue was white. Magnification: 100×.

**Figure 2 genes-15-01121-f002:**
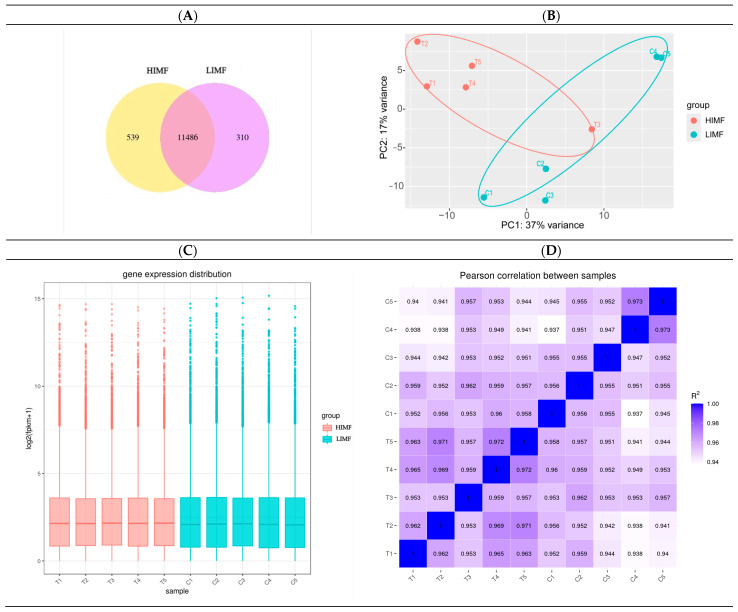
Quantitative analysis. (**A**) Sample co-expression Venn plot; (**B**) plot of results of principal component analysis (horizontal, first principal component; vertical, second principal component); (**C**) box plot of gene expression distribution of samples (horizontal coordinates are sample names, vertical coordinates are log_2_(FPKM + 1)); (**D**) heatmap of inter-sample correlations (horizontal and vertical coordinates are squared correlation coefficients for each sample).

**Figure 3 genes-15-01121-f003:**
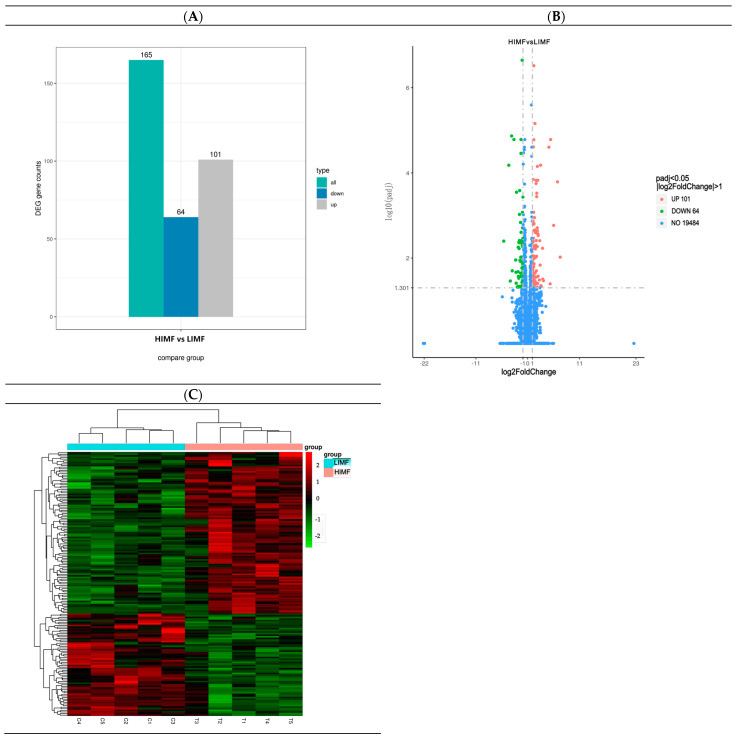
Map of differentially expressed genes. (**A**) Histogram of the number of differential genes counted in the differential comparison combinations; (**B**) differential gene volcano map (blue dashed lines indicate threshold lines for differential gene screening criteria; (**C**) heatmap of clustering of differentially expressed genes (the redder the color, the higher the expression; the greener the color, the lower the expression).

**Figure 4 genes-15-01121-f004:**
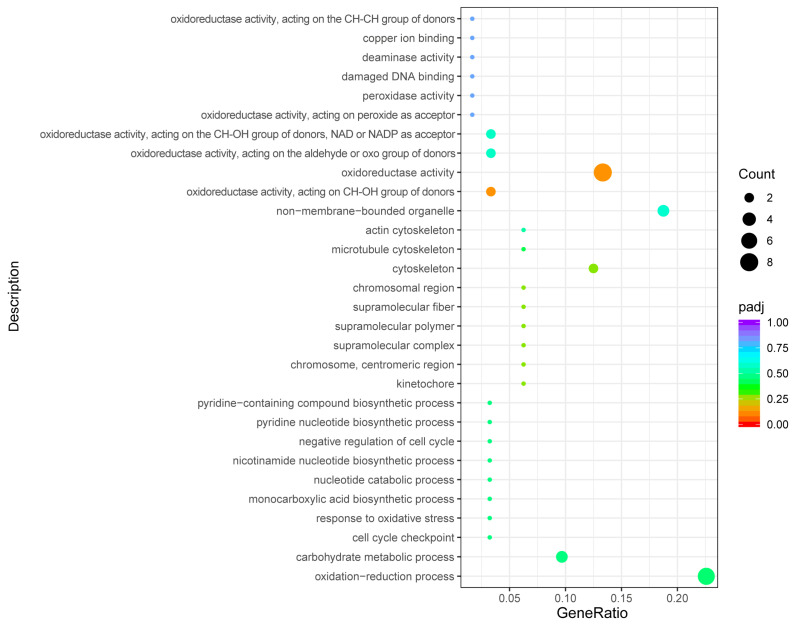
GO enrichment analysis. The color from red to purple represents the magnitude of the significance of the enrichment, and the size of the dots represents the number of enriched entries.

**Figure 5 genes-15-01121-f005:**
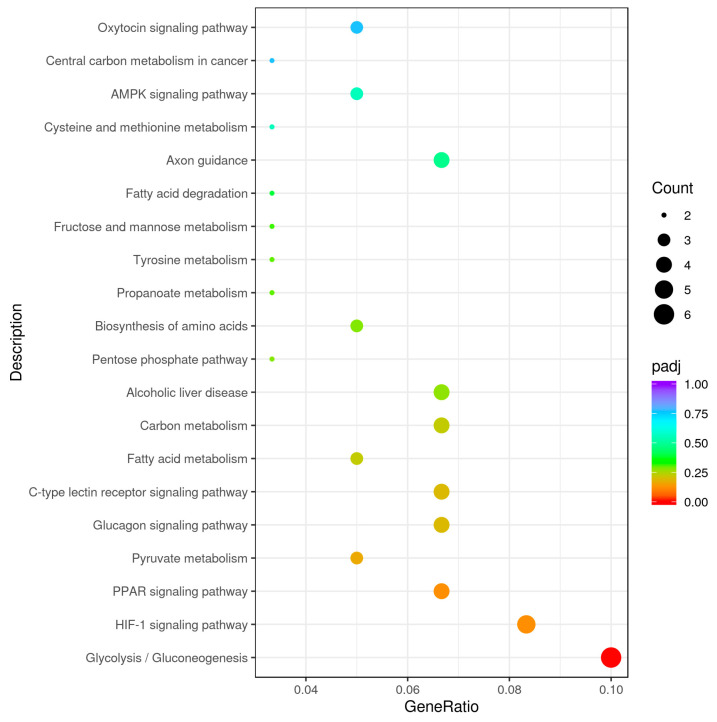
KEGG enrichment analysis. The KEGG pathway is annotated as the ratio of the number of differential genes to the total number of differential genes on the horizontal axis, and the KEGG pathway is annotated on the vertical axis.

**Figure 6 genes-15-01121-f006:**
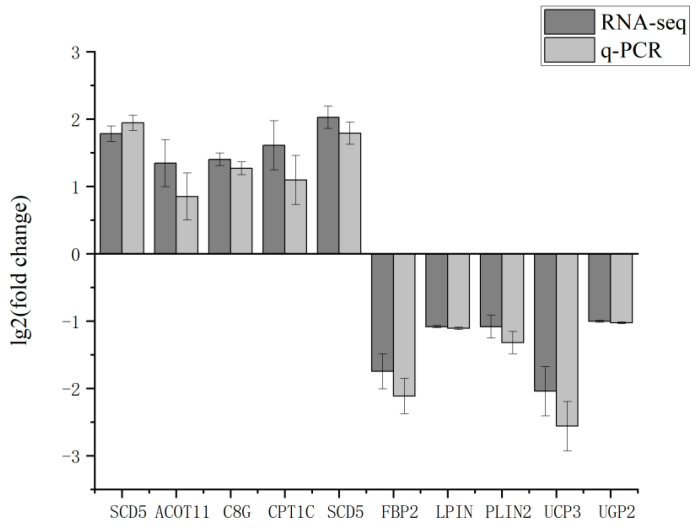
RT-qPCR verification of RNA-Seq results.

**Table 1 genes-15-01121-t001:** Primer sequences for RT-qPCR.

Gene	mRNA Accession Number	Primer Sequences (5′~3′)	Product Size
β-actin	NM_173979.3	Forward: AAGTACCCCATTGAGCACGG Reverse: TCCTTGATGTCACGGACGATTT	189 bp
SEC14L5	NM_001191259.3	Forward: CCACAAAGGCAAGATCCCCA Reverse: AGCTCAAGGACTGACACAGC	104 bp
C8G	NM_001110076.2	Forward: TCCCCTATCAGCACCATCCA Reverse: GCAGATTCCATCCAGCTTTCG	198 bp
CPT1C	XM_002695120.6	Forward: GGCTTCCGACCCTCACTGAC; Reverse: CAGAAACGGGAGAGATGCCTT	116 bp
ACOT11	NM_001103275.2	Forward: ATCCAGACTGTTGGAAATCACCT Reverse: CTCGCCATCTGCCATGTTGT	111 bp
SCD5	NM_001076945.1	Forward: TGGGTGCCATTGGTGAAGGT Reverse: CCCAGCCAACACATGAAGTC	124 bp
FBP2	NM_001046164.2	Forward: TTCATGCTTGACCCAGCTCTT Reverse: CCTCCATAGACAAGGGTGCG	230 bp
UCP3	NM_174210.1	Forward: CCCAACATCACGAGGAATGC Reverse: CAGGGGAAGTTGTCGGTGAG	107 bp
PLIN2	NM_173980.2	Forward: CTCCATTCCGCCTTCAACCT Reverse: ACGTGACTCAATGTGCTCAG	171 bp
UGP2	NM_174212.2	Forward: TGCGGATGTAAAGGGTGGGA Reverse: CATGCGCTTTTGGCACTTGA	83 bp
LPIN1	NM_001206156.2	Forward: TCTTCCCACTTCCACGCTTC Reverse: ATCCGCAGATTTGCTGACCA	90 bp

**Table 2 genes-15-01121-t002:** IMF content of the longissimus dorsi muscle in Xinjiang Brown Bulls.

Item ^1^	IMF Content (%)	Minimum (%)	Maximum (%)	*p*-Value
HIMF	6.996 ± 0.723 ^A^	5.863	7.824	0.002
LIMF	4.790 ± 0.783 ^B^	3.874	5.624	

^1^ HIMF: sample set of top 5 intramuscular fat contents, LIMF: the set of samples with the lowest 5 intramuscular fat contents. Results are expressed as mean ± standard deviation, *n* = 10. ^AB^ Values in the same column with different superscripts are extremely different (*p* < 0.01).

**Table 3 genes-15-01121-t003:** Summary of sample sequencing data quality.

Sample	Raw Reads	Raw Bases	Clean Reads	Clean Bases	Error Rate	Q20	Q30	GC pct
T1	43,246,440	6.49 G	42,439,552	6.37 G	0.03	97.81	93.85	51.02
T2	43,426,076	6.51 G	42,377,682	6.36 G	0.03	97.87	94.04	52.11
T3	41,992,442	6.3 G	41,202,940	6.18 G	0.02	98.03	94.47	52.4
T4	41,644,788	6.25 G	40,126,880	6.02 G	0.02	98.02	94.46	52.64
T5	48,098,022	7.21 G	46,682,976	7.0 G	0.03	97.9	94.16	51.8
C1	42,494,264	6.37 G	41,037,266	6.16 G	0.03	97.9	94.1	54.01
C2	43,497,090	6.52 G	42,302,406	6.35 G	0.02	98.03	94.44	51.74
C3	46,125,528	6.92 G	43,760,976	6.56 G	0.02	97.95	94.24	52.16
C4	48,924,916	7.34 G	47,173,470	7.08 G	0.02	98.08	94.58	50.79
C5	47,427,100	7.11 G	45,659,038	6.85 G	0.02	98.07	94.53	51.84

T1, T2, T3, T4 and T5 in the table are samples from the HIMF group; C1, C2, C3, C4 and C5 are samples from the LIMF group. The same as below.

**Table 4 genes-15-01121-t004:** Statistics on the comparison of the samples with the reference genome.

Sample	Total Reads	Total Map	Map Rate	Unique Map	Unique Map Rate	Multi Map	Multi Map Rate
T1	42,439,552	40,217,936	94.77%	39,141,933	92.23%	1,076,003	2.54%
T2	42,377,682	40,157,154	94.76%	38,944,182	91.9%	1,212,972	2.86%
T3	41,202,940	39,045,874	94.76%	37,817,486	91.78%	1,228,388	2.98%
T4	40,126,880	38,521,569	96.0%	37,350,423	93.08%	1,171,146	2.92%
T5	46,682,976	44,779,474	95.92%	43,397,543	92.96%	1,381,931	2.96%
C1	41,037,266	39,309,911	95.79%	38,015,521	92.64%	1,294,390	3.15%
C2	42,302,406	40,332,401	95.34%	39,075,620	92.37%	1,256,781	2.97%
C3	43,760,976	40,739,851	93.10%	39,403,210	90.04%	1,336,641	3.05%
C4	47,173,470	44,649,391	94.65%	43,182,346	91.54%	1,467,045	3.11%
C5	45,659,038	43,293,569	94.82%	41,735,269	91.41%	1,558,300	3.41%

**Table 5 genes-15-01121-t005:** KEGG enrichment pathway and differentially expressed genes.

KEGG ID	KEGG Description	*p*-Value	Upregulated Genes	Downregulated Genes
bta04152	AMPK signaling pathway	0.0710	*FBP2*; *SCD5*	*CPT1C*
bta04921	Oxytocin signaling pathway	0.1093	*RGS2*; *PTGS2*	*NFATC3*
bta05230	Central carbon metabolism in cancer	0.1063	*ENSBTAG00000032217*; *LDHB*	/
bta04360	Axon guidance	0.0550	*RND1*	*SSH2*; *UNC5A*; *NFATC3*
bta00270	Cysteine and methionine metabolism	0.0682	*ENSBTAG00000032217*; *LDHB*	/
bta00071	Fatty acid degradation	0.0417	*CPT1C*	*ENSBTAG00000052243*
bta00051	Fructose and mannose metabolism	0.0312	/	*FBP2*; *ALDOA*
bta00350	Tyrosine metabolism	0.0280	*ENSBTAG00000046264*	*TYRP1*
bta00640	Propanoate metabolism	0.0264	*ENSBTAG00000032217*; *LDHB*	/
bta01230	Biosynthesis of amino acids	0.0222	/	*ALDOA*; *ENSBTAG00000018554*
bta00030	Pentose phosphate pathway	0.0205	/	*FBP2*; *ALDOA*
bta04936	Alcoholic liver disease	0.0175	*SCD5*; *CPT1C*	*LPIN1*; *ENSBTAG00000052243*
bta01200	Carbon metabolism	0.0129	/	*FBP2*; *ALDOA*; *ENSBTAG00000018554*
bta01212	Fatty acid metabolism	0.0118	*SCD5*; *CPT1C*	*ENSBTAG00000052243*
bta04625	C-type lectin receptor signaling pathway	0.0083	*EGR3*; *PTGS2*	*CCL22*; *NFATC3*
bta04922	Glucagon signaling pathway	0.0072	*ENSBTAG00000032217*; *LDHB*; *CPT1C*	*FBP2*
bta00620	Pyruvate metabolism	0.0047	*ACOT11*; *ENSBTAG00000032217*; *LDHB*	/
bta03320	PPAR signaling pathway	0.0026	*SCD5*; *CPT1C*	*ENSBTAG00000052243*; *PLIN2*
bta04066	HIF-1 signaling pathway	0.0018	*ENSBTAG00000032217*; *LDHB*	*ALDOA*; *ENSBTAG00000018554*
bta00010	Glycolysis/gluconeogenesis	<0.0001	*ENSBTAG00000032217*; *LDHB*	*FBP2*; *ALDOA*; *ENSBTAG00000018554*

## Data Availability

We confirm that our experimental data are accurate, which supports the results and conclusions of this study.

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
