# Peer review of "Exploration of Genes Related to Intramuscular Fat Deposition in Xinjiang Brown Cattle"

_genes, 2024, doi:10.3390/genes15091121_

Round 1

Reviewer 1 Report

Comments and Suggestions for Authors

Dear authors, respectfully, it seems to me that this is a good manuscript. However, many points affect the quality of the manuscript. The main factor that affects the manuscript is the definition of an objective that would be directly correlated with all the other subtopics of the study.

Abstract: This abstract has relevant information, however distant of the objective of the study; also is generic. My suggestion is to add numbers to improve your description. For example, Samples with high IMF content showed 25% more activity in lipid pathways than samples with low IMF.

The objective describes a point that is not well answered by the conclusion. However, the material and methods describe a comparison of samples, so results on those comparisons are expected. There is a point that the authors tried to use to improve the study, but they left it unclear or the description is just confusing.

Add the experimental design, statistical analysis, correct number of experimental units.

Lines 11 and 13: There are two data on the number of experimental units. What is the correct value of the experimental units used in the study? 42 or 10? Or 10 samples for each of the 42 Longissimus?

Lines 16-17: Were there differences? Then describe what those differences were.

Line 19: GO enrich – what does GO stand for? A complete description is required before using only abbreviations. The same applies to other abbreviations such as on lines 22, 24 and throughout the text.

Lines 27-30: Is this the objective again? Why repeat it? Remove it.

Lines 30-32: This conclusion correlates with the overall objective; however, the experimental design and results are not aligned with this conclusion.

Keyword: My suggestion is to rewrite the keywords following the author's instructions. Also, the term "brown cattle" should be removed from the keywords. The keywords should be different from those in the title.

Introduction: Avoid informal writing style. Excellent, superior, inferior, good, strong, etc. These terms are relative and there is a chance that what is excellent for you may not be for others.

Lines 37-40: Very informal. Numbers are necessary to confirm and validate the description.

Lines 40-43: What studies? I don't understand the objective of these lines here.

Lines 51-54: Add the numbers of the bioenergetics.

Lines 61-62: How efficient is this technology? Add numbers about its reproducibility, efficiency, etc.

Lines 71-74: This objective is different from the abstract objective and I think you should think carefully and describe the specific objective of your study that relates to the title and other topics of the manuscript. As a suggestion: less is more.

I think adding the study hypothesis can help you improve your objective writing.

Lines 74-77: My issue with these lines is with the first words: “this process aims”. Rewrite it.

Material and methods: Why is the use of the longissimus of 42 animals described if only 10 were used as experimental units? Add the methods you followed to perform the analysis.

Line 80: Add the average body weight ± S.D.

Lines 83-84: Formulated taking into account what average daily weight gain?

Lines 92-95: Add the values ​​you considered high and low. In addition, there is a sub-topic for the IMF's determination, why repeat it here?

Line 97: How many grams?

Results:

Figure 2: To obtain these results, how is the data analyzed? This information must appear in the statistical topic.

Figure 2B: Those principal components explain 54% of the study, the this results is not representative. Also, describe what samples represent each principal component.

Figure 2D: What is the objective of the correlation between samples?

Discussion: The discussion topic clearly shows the lack of a specific objective of the study since the writing style of this is more of a review than a discussion.

The discussion should focus on explaining how the results were obtained. For this, add theories, hypotheses or statements about how you obtained your results, whether biologically, metabolically, physiologically, environmentally, etc. In the current situation, the discussion is a good general review (I do not recommend deleting any part of the discussion); however, you need to make a specific description of how the results were obtained.

Conclusion: Similar comment than for the discussion; however, I suggest to maintain lines 452-460 in the conclusion or add another sutopic denominated “future impliecations”. In this subtopic you will be able to answer the text described in the objective about how this information can be important for future knowledge of metabolism.

Lines 448-451: ok and? This is not a conclusion, this is a result description. Remove it or rewrite it.

Author Response

Thank you very much for your valuable comments on my manuscript “Exploration of Genes Related to Intramuscular Fat Deposition in Xinjiang Brown Bulls”. Your comments are very helpful to our research and will help to improve the quality of this manuscript. Based on your suggestions, we have revised the manuscript. Changes in the manuscript labeled in red are responses to you. Thank you again for reviewing the manuscript, and if you have any other comments or suggestions, we welcome your further guidance.

Comments 1: There are many areas in the manuscript that detract from the quality of the manuscript. The main factor that affects the manuscript is the definition of an objective that would be directly correlated with all the other subtopics of the study.

Response 1: Thank you for your questions about the manuscript. In this manuscript, our identification of objectives has been defined and is reflected in the abstract, introduction, and conclusion. Relevant corrections have been highlighted in red in the manuscript, and we would be grateful if you could correct any inappropriateness. See lines 32-37 (Abstract), 84-89 (Introduction), and 454 (Conclusion) in the revised manuscript for specific locations.

Comments 2: This abstract has relevant information, however distant of the objective of the study; also is generic. My suggestion is to add numbers to improve your description. For example, Samples with high IMF content showed 25% more activity in lipid pathways than samples with low IMF.

Response 2: Thank you for your questions about the abstract section of the manuscript, we have completed revisions to the abstract to make the information described in the abstract more specific and to clearly state the goals of the study. We have also added figures to make the information that needs to be presented in the manuscript clearer. See line 15 for the addition of the relevant figures.

Comments 3: The objective describes a point that is not well answered by the conclusion. However, the material and methods describe a comparison of samples, so results on those comparisons are expected. There is a point that the authors tried to use to improve the study, but they left it unclear or the description is just confusing.

Response 3: Thank you for pointing this out. We have rewritten the conclusions in relation to the research objectives to make them more relevant to the manuscript (lines 450-460). Sample comparisons were specifically described in Materials and Methods, making it clearer (lines 11-15).

Comments 4: Add the experimental design, statistical analysis, correct number of experimental units.

Response 4: Thank you for pointing out the problems that appeared in the abstract, which I have corrected individually in the manuscript (lines 10-20). If there is anything else that is not clearly expressed, I would also appreciate if you could give me any relevant comments.

Comments 5: There are two data on the number of experimental units. What is the correct value of the experimental units used in the study? 42 or 10? Or 10 samples for each of the 42 Longissimus? (Lines 11 and 13)

esponse 5: Thank you for pointing out the relevant issues. I apologize for not clearly expressing the experimental units used in the study in the original manuscript. In the new manuscript I have finalized the data on the number of experimental units as the collection of the longest dorsal muscles from 10 Xinjiang brown bulls.

Comments 6: Were there differences? Then describe what those differences were. (Lines 16-17)

Response 6: Thank you for your question here. there is indeed a difference in intramuscular fat content between the HIMF and LIMF groups, and I apologize for the lack of clarity in my presentation. Therefore, I made an adjustment in the manuscript, and changed “a significant difference in intramuscular fat content between the HIMF and LIMF groups” to “It was found that the fat richness of the HIMF group was better than that of the LIMF group.”

Comments 7: GO enrich – what does GO stand for? A complete description is required before using only abbreviations. The same applies to other abbreviations such as on lines 22, 24 and throughout the text. (Line 19)

Response 7: Thank you for your opinionated guidance, I couldn't agree with you more. I have provided a complete description of the abbreviations such as GO, KEGG, etc., which appear for the first time in the manuscript. For example, Gene Ontology (GO) enrichment analysis, Kyoto Encyclopedia of Genes and Genomes (KEGG) enrichment analysis, Peroxisome Proliferator-Activated Receptors (PPAR) signalling pathway, intramuscular fat (IMF).

Comments 8: Is this the objective again? Why repeat it? Remove it. (Lines 27-30)

Response 8: I agree with the comments you gave. I have removed this part of the original manuscript. Thank you again for your guidance!

Comments 9: This conclusion correlates with the overall objective; however, the experimental design and results are not aligned with this conclusion. (Lines 30-32)

Response 9: Thank you for the valuable comments you have given. I have rewritten the sentences in lines 30-32 of the original manuscript to make the conclusions more closely related to the experimental design and results. The revised sentence reads “Therefore, we hypothesized that SCD5, CPT1C, and FBP2 were the key genes responsible for the significant differences in intramuscular fat content of the longissimus dorsi muscles in a population of Xinjiang Brown Bulls. We expect that these findings will provide fundamental support for subsequent studies exploring key genes affecting fat deposition characteristics in Xinjiang Brown Cattle.”

Comments 10: My suggestion is to rewrite the keywords following the author's instructions. Also, the term "brown cattle" should be removed from the keywords. The keywords should be different from those in the title.

Response 10: Thank you very much for the advice you have provided and I agree with you. I have made changes to the keywords in the manuscript and deleted “brown cattle”. The revised keywords are longissimus dorsi muscle, intramuscular fat, differentially expressed gene, transcriptome sequencing technology and metabolic pathway. I would be grateful if you could point out any corrections.

Comments 11: Avoid informal writing style. Excellent, superior, inferior, good, strong, etc. These terms are relative and there is a chance that what is excellent for you may not be for others.

Response 11: Thank you very much for your point of view! I also realized the inappropriate terminology that appeared when I wrote the introduction. Therefore I have examined the manuscript and corrected it to avoid terms such as excellent, good and strong. Some descriptions in the manuscript, such as “Xinjiang Brown Cattle inherited the strong adaptability of its female Kazakh Cattle (heritability 9.88%) and the better production performance( beef and dairy) of its male Swiss Brown Cattle (heritability 90.12%)”, are based on the previous research. Finally, if there are still problems with my presentation in the manuscript, I hope you will give your valuable comments.

Comments 12: Very informal. Numbers are necessary to confirm and validate the description. (Lines 37-40)

Response 12: I agree with your comments and have adopted your suggestion that “numbers are necessary to confirm and validate descriptions”. In the manuscript, I have rewritten the introductory sentence to describe it as follows: Xinjiang Brown Cattle inherited the strong adaptability of its female Kazakh Cattle (heritability 9.88%) and the better production performance( beef and dairy) of its male Swiss Brown Cattle (heritability 90.12%).

Comments 13: What studies? I don't understand the objective of these lines here. (Lines 40-43)

Response 13: I appreciate the confusion you expressed about this sentence. I really didn't make it clear what study showed that “under consistent feeding conditions, the Xinjiang Brown cattle breed showed a diversity of meat quality characteristics, including differences in intramuscular fat (IMF) content.” The results of the Therefore, I deleted the sentence and then rewrote it based on the previous article. Specifically, it was as follows: At present, through nutritional regulation and other technical means, the carcass quality grade of Xinjiang Brown Cattle has generally reached the level of R grade above (Which was refer to European Union( EU) beef cattle carcass quality grade standdard ). Previous studies have shown that Xinjiang Brown Steers ( 10-12 month age) the ribeye area was 106.06㎝², fat cover was 3.5%, the marbling score was grade 3 to 4 and the fat color score was grade 2 ( which were refer to NY/T 676-2010 Beef Grade Specification). Therefore, Xinjiang Brown Cattle can be the main target of high-grade beef production.

Comments 14: Add the numbers of the bioenergetics. (Lines 51-54)

Response 14: I appreciate and am happy to take the advice you give. It is true that the sentence in the original manuscript lacked a bioenergetic figure to support it, but I apologize for not being able to find one in my review of the literature. However, I have rewritten the relevant part of the manuscript by reviewing other literature, i.e., “Yan et al. reported that the IMF content of Xinjiang Brown Steer ( 6.96 - 7.50%) was higher than Angus Steer (6.2%).” If you find anything inappropriate please point it out and I would be happy to have your guidance.

Comments 15: How efficient is this technology? Add numbers about its reproducibility, efficiency, etc. (Lines 61-62)

Response 15: Thank you very much for your valuable comments. I have added some values to the manuscript to demonstrate the accuracy of the RNA-Seq technique. Namely, “RNA-Seq technology is capable of achieving read lengths of 2×150bp, and it has the advantage of high sequencing throughput and high accuracy (>99%) for precise quantification” If my statement is not clear and accurate please give me your valuable advice, I will be grateful.

Comments 16: This objective is different from the abstract objective and I think you should think carefully and describe the specific objective of your study that relates to the title and other topics of the manuscript. As a suggestion: less is more. (Lines 71-74)

Response 16: I appreciate and accept the advice you give. That is why I have rearranged the narrative of the study objectives in the manuscript. That is, “We used transcriptome sequencing to analyze the differentially expressed genes in the longissimus dorsi muscle of Xinjiang Brown Cattle, and explored the genes and pathways closely related to intramuscular fat deposition in conjunction with existing studies to gain insight into the key genes that regulate intramuscular fat deposition.”.

Comments 17: I think adding the study hypothesis can help you improve your objective writing.

Response 17: I appreciate and am happy to take the guidance you give. So I have added the relevant research hypotheses in the manuscript. That is, “Therefore, we hypothesized that certain genes affect intramuscular fat deposition in Xinjiang Brown Cattle.” . If you find any inappropriateness please raise it immediately and I will be glad to accept your guidance.

Comments 18: My issue with these lines is with the first words: “this process aims”. Rewrite it. (Lines 74-77)

Response 18: I take the advice you have given. I have rewritten the relevant part of the sentence in the manuscript. That is, “We used transcriptome sequencing to analyze the differentially expressed genes in the longissimus dorsi muscle of Xinjiang Brown Cattle, and explored the genes and pathways closely related to intramuscular fat deposition in conjunction with existing studies to gain insight into the key genes that regulate intramuscular fat deposition.” Thank you again for your guidance.

Comments 19: Why is the use of the longissimus of 42 animals described if only 10 were used as experimental units? Add the methods you followed to perform the analysis.

Response 19: I apologize for the lack of clarity here and thank you for pointing out the problems that have arisen here. I have redrawn it in the manuscript. Of course I need to explain here that I randomly selected 10 of the 42 Xinjiang Brown bulls for slaughter and then took the longest dorsal muscles of these 10 Xinjiang Brown bulls. So the test samples were 10 dorsal longest muscle samples. That is, “In this experiment, 10 healthy 26-month-old Xinjiang Brown Bulls (653.21±67.27) with the same genetic background and kept under the same feeding conditions were randomly selected at the Xinjiang Yili New Brown Breeding Farm. ”

Comments 20: Add the average body weight ± S.D. (Line 80)

Response 20: I accept the advice you gave, it was indeed an oversight on my part and for that I am deeply sorry. I have added the average weight ± S.D. of brown bulls in Xinjiang to the manuscript. That is “10 healthy 26-month-old Xinjiang Brown Bulls (653.21±67.27)”.

Comments 21: Formulated taking into account what average daily weight gain? (Lines 83-84)

Response 21: Thank you for the guidance you have given. Regarding your query “Does it take into account what the average daily weight is?”, I am not able to give you a definite reply. This is because I have not been fully involved in documenting each weighing of my test subject Xinjiang Brown bulls during their growth. I only weighed 10 selected Xinjiang Brown bulls. However, I can assure you that all test animals were at the same nutritional level and that the diets were formulated according to the NRC (National Research Council) Nutrient Requirements for Beef Cattle. See “Academy of Sciences, Engineering Academy, and Institute of Medicine of the National Academies. Nutrient Requirements of Beef Cattle: Eighth Revised Edition [M]. Beijing: Science Press, 2018. ISBN: 9787030572905”. If my explanation is not clear enough please point it out and I'll be happy to take your guidance.

Comments 22: Add the values ​​you considered high and low. In addition, there is a sub-topic for the IMF's determination, why repeat it here? (Lines 92-95)

Response 22: Thank you very much for the valuable advice you have given! For the determination of HIMF and LIMF groups I redefined them in lines 104-110 in the manuscript, i.e.: We determined the intramuscular fat (IMF) content of the longissimus dorsi muscle of 10 Xinjiang Brown Bulls using the Soxhlet extraction method, and subsequently ranked the samples according to the measured content from highest to lowest. On this basis, the top five samples with the highest IMF content were selected as the HIMF group, while the remaining samples were categorized as the LMIF group. If any of this is not clear to you please point it out and I will be more than happy to take your advice.

Comments 23: How many grams? (Line 97)

Response 23: You have asked questions that I am happy to answer. For the longest dorsal muscle samples in the HIMF and LIMF groups, we split the collected samples into two. One portion was taken 20-50 g for making paraffin sections for morphological observation; the other portion was taken 2-4 g for transcriptome sequencing technology to analyze the gene expression differences of the longest dorsal muscle between the two groups.

Comments 24: To obtain these results, how is the data analyzed? This information must appear in the statistical topic. (Figure 2)

Response 24: Thank you very much for your suggestions. For the results of how the figure 2 part is analyzed and data processing I did not show in the statistics thread is my oversight. So I added a description of the relevant part in the manuscript, namely “Gene quantification was performed using the default parameters of the feature Counts (1.5.0-p3) software. R (Version 3.5.0) was used to plot heatmaps of correlation between samples (cor function package, method = person; ggplot2), box plots of the distribution of gene expression in the samples (ggplot2 package), plots of the results of the principal component analysis (ggplot2 package), and Venn Diagrams of co-expression (Venn Diagram package).” . Please correct me if there are any problems with my description and I will be more than happy to make changes.

Comments 25: Those principal components explain 54% of the study, the this results is not representative. Also, describe what samples represent each principal component. (Figure 2B)

Response 25: Thank you for your valuable input. Regarding your question “the principal components explain 54% of the study, the result is not representative”, I have checked other literature (Chen, Q.; Xu, L.; Zhang, M.; Zhang, T.; Yan, M.; Zhai, M.,; Huang, X. Whole genome resequencing reveals the genetic contribution of Kazakh and Swiss Brown Cattle to a population of Xinjiang Brown Cattle. Gene 2022, 839, 146725. https://doi.org/10.1016/j.gene.2022.146725) found that even though my principal component analysis is only 54%, it does not mean it lacks representativeness. If the principal component analysis shows that the first few principal components together explain 54% of the variability in the data, this means that these principal components capture most of the important information in the data, but there is still 46% of the variability that is not explained by these principal components. This does not mean that the unexplained portions are not important, just that they are not well represented under the currently selected principal components. In addition, I have made a change to the graph where each principal component represents that sample (Below line 267). I apologize for the oversight, if you have any further questions please feel free to ask and I will be happy to address them.

Comments 26: What is the objective of the correlation between samples? (Figure 2D)

Response 26: Thank you very much for your valuable input. Regarding the question “What is the purpose of Figure 2D inter-sample correlation?” My answer is that a heatmap of inter-sample correlation can not only show the correlation between data and the strength of the correlation between different samples in an intuitive and compelling way, but also assist researchers to perform cluster analysis and improve the interpretation of the data and assist in decision making. Meanwhile, the correlation of gene expression levels between samples is an important indicator to test the reliability of the experiment and whether the sample selection is reasonable. The closer the correlation coefficient is to 1, the higher the similarity of expression patterns between samples. If you don't understand any part of my explanation, please point it out, I will be very happy to answer.

Comments 27: The discussion topic clearly shows the lack of a specific objective of the study since the writing style of this is more of a review than a discussion. 

The discussion should focus on explaining how the results were obtained. For this, add theories, hypotheses or statements about how you obtained your results, whether biologically, metabolically, physiologically, environmentally, etc. In the current situation, the discussion is a good general review (I do not recommend deleting any part of the discussion); however, you need to make a specific description of how the results were obtained.

Response 27: I'm happy to take your comments and make changes accordingly. In the Discussion section of the manuscript, I have followed your suggestions that “The discussion title clearly indicates the lack of a specific research objective” and “The discussion should focus on explaining how the results were obtained. This can be done by adding theories, hypotheses, or statements about how you obtained the results biologically, metabolically, physiologically, environmentally, etc.” The manuscript was revised based on two comments. Since I tweaked and revised the discussion section of the manuscript as a whole, I will not describe it here. The specific revisions are found in lines 329-448 of the manuscript. I apologize for any inconvenience this may cause you, or if you have any questions, please feel free to ask and I will be happy to answer them.

Comments 28: Similar comment than for the discussion; however, I suggest to maintain lines 452-460 in the conclusion or add another sutopic denominated “future impliecations”. In this subtopic you will be able to answer the text described in the objective about how this information can be important for future knowledge of metabolism. (Conclusion)

Response 28: I am very grateful and happy to receive your valuable comments. I have redescribed the conclusions to make them more in line with the manuscript. I have also taken on board your suggestion to retain lines 452-460 or to add a subheading entitled “Future Implications”. The conclusion in the manuscript (lines 450-460) is: Overall, our study indicates that SCD5 and CPT1C genes (related to fatty acid metabolism and PPAR signaling pathway) can indirectly or directly promote intramuscular fat deposition, while the FBP2 gene (involved in glycolysis/glycogenesis and AMPK signaling pathway) can influence adipose lipid metabolism through glucose metabolism, suggesting that SCD5, CPT1C, and FBP2 are differentially expressed genes that participate in regulating lipid metabolism-related processes. Therefore, it is necessary to further investigate the potential mechanisms by which SCD5, CPT1C and FBP2 genes regulate intramuscular lipid content in the longissimus dorsi muscle of Xinjiang Brown Cattle. Despite the limitations of this study, these results provide a reference for exploring the key regulatory genes affecting the fat deposition process in cattle. I would also appreciate it if you could point out any changes or adjustments that need to be made to the manuscript.

Comments 29: ok and? This is not a conclusion, this is a result description. Remove it or rewrite it. (Lines 448-451)

Response 29: Thanks for your comments on the conclusion, which I am very pleased to accept. In the manuscript I have deleted this part of the formulation and rewritten the part of the conclusion (see lines 450-457). If you have any suggestions, please let me know and I will be happy to make changes.

Reviewer 2 Report

Comments and Suggestions for Authors

The presented manuscript on the topic: "Exploration of Genes Related to Intramuscular Fat Deposition in Xinjiang Brown Cattle" is extremely interesting because it examines the genes responsible for the accumulation of fat in the muscles of a breed of cattle indigenous to the Republic of China. In this aspect, it is very possible that these local breeds under the conditions of climate change will prove to be a valuable genetic resource not only for China but also for the whole world.

There are some notes to the authors:

Are the animals used in the experiment of the same sex, and if not, how many are male and female, and have the males been castrated, because this significantly affects the accumulation of fat.

On what basis are high and low fat-accumulators divided - body mass, body condition score, please clarify?

Has the composition of fatty acids in the muscles been studied, because it is very important not only the quantity but also the ratio of fatty acids and the resulting health lipid indices?

Author Response

Thank you very much for your valuable comments on my manuscript “Exploration of Genes Related to Intramuscular Fat Deposition in Xinjiang Brown Bulls”. Your comments are very helpful to our research and will help to improve the quality of this manuscript. Based on your suggestions, we have revised the manuscript. Changes in the manuscript labeled in blue are responses to you. Thank you again for reviewing the manuscript, and if you have any other comments or suggestions, we welcome your further guidance.

Comments 1: Are the animals used in the experiment of the same sex, and if not, how many are male and female, and have the males been castrated, because this significantly affects the accumulation of fat.

Response 1: I thank you very much for pointing out the problems on the test animals, this comment is invaluable. I deeply apologize for not making it clear in my manuscript whether the test animals were homosexual and castrated or not. I have explained this in the manuscript and selected male, uncastrated Xinjiang brown cattle ( Xinjiang Brown Bulls in the manuscript). I hope you understand what I have conveyed, and if you have any questions, please point them out to me and I will be happy to answer them.

Comments 2: On what basis are high and low fat-accumulators divided - body mass, body condition score, please clarify?

Response 2: The comments you have made are invaluable. I apologize for not clearly expressing “how to classify high fat accumulators and low fat accumulators” in my manuscript. Therefore, I made a specific statement in the manuscript, that is, “The longissimus dorsi muscles of 10 Xinjiang Brown Bulls were selected under the same feeding conditions. The intramuscular fat content of muscle samples were determined by Soxhlet extraction method, from which 5 samples with high intramuscular fat content (HIMF group) and 5 samples with low intramuscular fat content (LIMF group) were selected.” If you still have any ambiguities please also point them out and I will be happy to make changes.

Comments 3: Has the composition of fatty acids in the muscles been studied, because it is very important not only the quantity but also the ratio of fatty acids and the resulting health lipid indices?

Response 3: I am very interested in what you are proposing. Unfortunately I have not conducted a study of the fatty acid composition of the samples in this test. But this is a very good tip, unfortunately I don't have any extra samples to test the fatty acid composition of the muscles again. Thank you again for your input and I apologize for not having the relevant data.

Round 2

Reviewer 1 Report

Comments and Suggestions for Authors

Dear authors, reviewers make suggestions with the aim of improving the manuscript; however, it is the responsibility of the authors to accept them in full, accept them partially or reject them. I am satisfied with the responses; however, I have one more request, to add the deleted revision described in the previously revised manuscript and merge it with the current discussion.

Author Response

Dear reviewer,

Thank you very much for your valuable comments on my manuscript “Exploration of genes associated with intramuscular fat deposition in Xinjiang brown cattle”. On this occasion, I have revised the manuscript according to the suggestions of reviewer 1, and marked the text in steel blue (see the Discussion section, lines 334-484, for details)

Comments 1: Dear authors, reviewers make suggestions with the aim of improving the manuscript; however, it is the responsibility of the authors to accept them in full, accept them partially or reject them. I am satisfied with the responses; however, I have one more request, to add the deleted revision described in the previously revised manuscript and merge it with the current discussion.

Response 1: I am pleased to receive your feedback and I am happy to make changes accordingly based on your comments. In the “Discussion” section of the manuscript, I accept your suggestion that “the deleted revisions be added to the previously revised manuscript and merged with the current discussion.” . Since there are more parts of the Discussion section that have been revised and adjusted, I will not repeat them here. See lines 329-484 for specific changes. At the same time, I added a section on “Future Perspectives” to the Discussion section (lines 470-484).

I apologize for any inconvenience; if you have any questions, please feel free to ask and I will be happy to answer them.

Your comments are very helpful to our research and help to improve the quality of this manuscript. Thank you again for reviewing the manuscript, and if you have any other comments or suggestions, we welcome your further guidance.

Reviewer 2 Report

Comments and Suggestions for Authors

I accept the manuscript without comment and recommend that it be published as it stands.

Author Response

Dear reviewer,

It is a great honor to receive your response to my manuscript “Discussion of genes associated with intramuscular fat deposition in Xinjiang brown cattle”. I am very happy that my manuscript is recognized by you, and I would like to express my great appreciation.

Thank you again for your approval.

Have a nice day!

Yu Gao